# Cluster Analysis in Diabetes Research: A Systematic Review Enhanced by a Cross-Sectional Study

**DOI:** 10.3390/jcm14103588

**Published:** 2025-05-21

**Authors:** Binura Taurbekova, Radmir Sarsenov, Muhammad M. Yaqoob, Kuralay Atageldiyeva, Yuliya Semenova, Siamac Fazli, Andrey Starodubov, Akmaral Angalieva, Antonio Sarria-Santamera

**Affiliations:** 1Department of Biomedical Sciences, School of Medicine, Nazarbayev University, 5/1 Kerey and Zhanibek Khandar Str., Astana 010000, Kazakhstan; antonio.sarria@nu.edu.kz; 2Department of Biology, School of Sciences and Humanities, Nazarbayev University, 53 Kabanbay Batyr Ave., Astana 010000, Kazakhstan; radmir.sarsenov@nu.edu.kz; 3Department of Renal Medicine and Transplantation, The Royal London Hospital, Barts Health NHS Trust, Whitechapel Road, London E1 1BB, UK; m.m.yaqoob@qmul.ac.uk; 4William Harvey Research Institute, Faculty of Medicine & Dentistry, Queen Mary University of London, Charterhouse Square, London EC1M 6BQ, UK; 5Department of Medicine, School of Medicine, Nazarbayev University, 5/1 Kerey and Zhanibek Khandar Str., Astana 010000, Kazakhstan; kuralay.atageldiyeva@nu.edu.kz; 6Department of Surgery, School of Medicine, Nazarbayev University, 5/1 Kerey and Zhanibek Khandar Str., Astana 010000, Kazakhstan; yuliya.semenova@nu.edu.kz; 7Department of Computer Science, School of Engineering and Digital Sciences, Nazarbayev University, 53 Kabanbay Batyr Ave., Astana 010000, Kazakhstan; siamac.fazli@nu.edu.kz; 8«B.B.NURA» Hospitals Group, Office 815, 33/1 Mangilik El Str., Astana 010000, Kazakhstan; a.starodubov@bbnura.kz; 9Women’s Health Department, City Multidisciplinary Hospital No. 2, 6 Turar Ryskulov Str., Astana 010000, Kazakhstan; akmaralangalieva@gmail.com

**Keywords:** diabetes mellitus, cluster analysis, phenotype, classification, diabetic complication

## Abstract

**Background:** Diabetes mellitus is a heterogeneous metabolic disorder that poses substantial challenges in the management of patients with diabetes. Emerging research underscores the potential of unsupervised cluster analysis as a promising methodological approach for unraveling the complex heterogeneity of diabetes mellitus. This systematic review evaluated the effectiveness of unsupervised cluster analysis in identifying diabetes phenotypes, elucidating the risks of diabetes-related complications, and distinguishing treatment responses. **Methods:** We searched MEDLINE Complete, PubMed, and Web of Science and reviewed forty-one relevant studies. Additionally, we conducted a cross-sectional study using K-means cluster analysis of real-world clinical data from 558 patients with diabetes. **Results:** A key finding was the consistent reproducibility of the five clusters across diverse populations, encompassing various patient origins and ethnic backgrounds. MOD and MARD were the most prevalent clusters, while SAID was the least prevalent. Subgroup analysis stratified by ethnic group indicated a higher prevalence of SIDD among individuals of Asian descent than among other ethnic groups. These clusters shared similar phenotypic traits and risk profiles for complications, with some variations in their distribution and key clinical variables. Notably, the SIRD subtype was associated with a wide spectrum of kidney-related clinical presentations. Alternative clustering techniques may reveal additional clinically relevant diabetes subtypes. Our cross-sectional study identified five subgroups, each with distinct profiles of glycemic control, lipid metabolism, blood pressure, and renal function. **Conclusions:** Overall, the results suggest that unsupervised cluster analysis holds promise for revealing clinically meaningful subgroups with distinct characteristics, complication risks, and treatment responses that may remain undetected using conventional approaches.

## 1. Introduction

Diabetes mellitus has become one of the fastest-growing healthcare challenges of the 21st century [1]. According to recent statistics from the International Diabetes Federation (IDF), the prevalence of diabetes has increased sharply worldwide, with 537 million adults affected in 2021 [1]. Given the heterogeneity in the clinical manifestations, trajectories, and outcomes of diabetes [2], the conventional classification of diabetes into primarily type 1 (T1D) and type 2 diabetes (T2D) is widely considered insufficient to capture the complexity of the disease [3]. Subdividing diabetes into more homogeneous subgroups may enhance risk stratification for complications and refine therapeutic approaches, potentially improving treatment efficacy and patient prognosis [2].

Recently, Ahlqvist et al. [4] made an unprecedented contribution to the identification of distinct diabetes phenotypes using data-driven cluster analysis. The ANDIS (All New Diabetics in Scania) cohort, which consisted of individuals with newly diagnosed diabetes, was assigned to clusters based on phenotypic similarity, as defined by six variables. The included model variables reflect important risk factors, illuminate key pathophysiological features, and can be relatively easily determined in routine clinical practice: glutamic acid decarboxylase antibodies (GADA), age at onset of diabetes, body mass index (BMI), hemoglobin A1c (HbA1c), homeostasis model assessment (HOMA) 2 estimates of β-cell function (HOMA2-B), and insulin resistance (HOMA2-IR). Using K-means and hierarchical clustering, five phenotypic diabetes subtypes were identified and named based on their distinctive features: Severe Autoimmune Diabetes (SAID), Severe Insulin-Deficient Diabetes (SIDD), Severe Insulin-Resistant Diabetes (SIRD), Mild Obesity-Related Diabetes (MOD), and Mild Age-Related Diabetes (MARD).

This innovative clustering strategy for diabetes subphenotyping has generated enormous scientific interest. Over the past few years, the clustering technique has been applied to various populations of people with diabetes using either a similar method as in the original study [4] or diverse clustering algorithms and model variables. Overall, five subtypes have been widely replicated across studies involving participants of different ethnic backgrounds [5]. These clusters differ in clinical presentation: SAID and SIDD are characterized by early onset, low BMI, insulin deficiency, and poor glycemic control, with SAID defined by GADA positivity; SIRD features severe insulin resistance and high BMI; MOD includes younger individuals with obesity and moderate insulin resistance; and MARD comprises older participants with relatively mild metabolic disturbances [5]. A prior systematic review indicated that some studies were able to reproduce the same clusters with similar traits, while other research revealed notable discrepancies in both the distribution of clusters and the central tendencies of the variables utilized in the algorithmic models [5]. Research that applied other clustering techniques and/or variables yielded notable distinctions [5]. It remains uncertain whether the noted discrepancy is linked to variations in the dominant pathophysiological features of diabetes among different populations or arises from the selection of a particular clustering approach.

A recent whole-genome sequencing study of healthy ethnic Kazakhs confirmed that this population has a unique genetic profile characterized by substantial admixture from both European and Asian ancestral lineages [6]. This high level of admixture is typical of Central Asian populations and reflects the region’s historical role as a crossroads for migration and trade [6]. The study also identified genetic variants associated with metabolic traits, including T2D and hypertension [6]. These findings underscore the importance of investigating disease patterns in this region, particularly in the context of metabolic disorders common in Central Asia.

Building on this background, this systematic review aimed to extend previous research efforts [5] and evaluate whether unsupervised cluster analysis could interpret complex interactions within data and identify distinct subgroups with clinically significant differences in disease trajectories, risk of diabetes-related complications, and treatment response. To support this aim, we performed a cross-sectional study using real-world clinical data from Kazakhstan, a Central Asian country with a unique genetic profile and environmental background, to further assess the efficacy of unsupervised cluster analysis in identifying distinct diabetes phenotypes.

## 2. Materials and Methods

### 2.1. Systematic Review

We conducted a systematic review following the Preferred Reporting Items for Systematic Reviews and Meta-Analyses (PRISMA) guidelines [7] and the pre-established PROSPERO research protocol (CRD42024609962).

#### 2.1.1. Search Strategy

Three English electronic bibliographic databases, namely MEDLINE Complete, PubMed, and Web of Science, were searched to retrieve potentially relevant publications that reported the results of subclassification of diabetes using an unsupervised clustering approach. The comprehensive search strategies included the terms «diabetes mellitus» and «cluster analysis» using various combinations of medical subject heading (MeSH) terms and free text words. The full database search strategies are detailed in Table 1. To supplement the electronic database search, the reference lists of the included studies and relevant review articles were screened to identify other potentially eligible publications. Following a previous systematic review [5], the literature search strategy included searching for English-language studies published between August 2020 and August 2024. The retrieved publications were organized and managed using Zotero reference management software (version 6.0.37).

#### 2.1.2. Selection Criteria and Data Extraction

The inclusion criteria were as follows: (1) studies that included patients diagnosed with diabetes, regardless of type; (2) implementation of a subclassification of diabetes using an unsupervised clustering approach; and (3) full-text articles that included research participants without age restrictions. Non-original works, including reviews and meeting abstracts, were excluded. Studies that involved research participants with diabetes and certain comorbidities, such as established cardiovascular disease, or those with specific diabetes-related complications or on a particular treatment, were excluded to preserve comparability and consistency within clusters.

After removing duplicate records, the remaining studies were independently reviewed and extracted by two investigators (B.T. and A.A.). All discrepancies were resolved through discussion to achieve consensus. In cases where consensus was not reached, a senior reviewer (A.S.-S.) was consulted and intervened to resolve inconsistencies. The following information was extracted from each eligible study: first author’s name, year of publication, ethnicity/geographic region, study design, data source, sample size and its characteristics, diabetes diagnostic criteria, clustering methods and dimensionality reduction techniques, methods for determining the number of clusters, variables for cluster analysis, and identified clusters and their characteristics.

#### 2.1.3. Statistical Analysis

Data analysis was conducted using STATA software (version MP17.0). The pooled prevalence of clusters was estimated along with the corresponding 95% confidence interval (95% CI) after Freeman-Tukey Double Arcsine Transformation. Heterogeneity and inconsistency were measured using the chi-square test (Cochrane Q statistic) and inconsistency index (I^2^). Given the significant heterogeneity between studies, a random-effects model (REM) was employed to calculate the overall prevalence. Subgroup analyses were conducted based on ethnic groups to investigate potential sources of heterogeneity and variations in prevalence. Studies involving participants of multiple ethnicities were excluded from the subgroup analysis to maintain comparability across groups.

#### 2.1.4. Quality Assessment

The methodological quality of each study included in this systematic review was independently evaluated by two researchers (B.T. and R.S.) using the National Heart, Lung, and Blood Institute (NHLBI) tool for cohorts [8] and the Joanna Briggs Institute (JBI) Critical Appraisal Tool for cross-sectional studies [9]. Any disagreements between the two reviewers were resolved by a senior reviewer (A.S.-S.). Additionally, we examined the methodological quality of the randomized controlled trials (RCTs) used as data sources for several studies. These studies not only performed cluster analyses but also assessed the intervention effects within the clusters. The evaluation was performed using the Revised Cochrane Risk-of-Bias Tool for RCTs (RoB 2) [10].

### 2.2. Cross-Sectional Study

#### 2.2.1. Study Population and Design

Following the decision of the Nazarbayev University School of Medicine Institutional Research Ethics Committee on 14 December 2022, the cross-sectional study (NUSOM-IREC 2022DEC#06) was exempted from the Research Ethics Review.

This cross-sectional study was conducted between December 2022 and January 2023. Data were collected from four outpatient clinics in Astana, Kazakhstan. We analyzed de-identified electronic health record (EHR) data from 558 patients aged 18 years and older, diagnosed with T2D or T1D between 2019 and 2022. A total of 77.5% of the study participants self-identified as ethnic Kazakhs and 20.6% as Russian. The remaining participants were Korean, German, Chinese, Ingush, Armenian, and Tatar. Women comprised 56.5% of the study population.

Diabetes diagnosis was defined using the International Classification of Diseases (ICD) codes. The age at diagnosis refers to the age of the patients when they were first diagnosed with diabetes, as documented in the EHR. All other relevant EHR data were extracted at the time closest to the diabetes diagnosis.

The inclusion criteria were a documented diagnosis of diabetes (T1D or T2D) and the availability of complete clinical or laboratory data required for cluster assignment. The exclusion criteria included secondary forms of diabetes (e.g., steroid-induced or due to pancreatic disorders), gestational diabetes, and age under 18 years.

#### 2.2.2. Cluster Analysis

We employed K-means cluster analysis to identify subgroups of diabetes based on a set of variables. Specifically, nine distinct variables were selected for clustering: age at diagnosis, BMI, systolic blood pressure (SBP), diastolic blood pressure (DBP), HbA1c, fasting plasma glucose (FPG), total cholesterol (TC), low-density lipoprotein cholesterol (LDL-C), and estimated glomerular filtration rate (eGFR). The eGFR was calculated using the Chronic Kidney Disease Epidemiology Collaboration (CKD-EPI) creatinine equation. The values were scaled to a range between 0 and 1 using the MinMaxScaler function from the scikit-learn Python machine learning library (version 3.10.8). This process ensures that all variables are normalized and contribute equally to the clustering process without introducing any bias. We applied the Elbow method to determine the optimal number of clusters by assessing the values of k ranging from 2 to 20. The within-cluster sum of squares (WCSS) was computed and plotted against the corresponding k values. The optimal number of clusters was identified at the point where increasing k no longer led to improvements in WCSS, with five clusters being determined as optimal. To validate this, the Silhouette width method was used, as it measures how well each data point fits within its assigned cluster (cohesion) compared to its closest alternative cluster (separation). The findings are visually represented by a graph created using Flourish Studio (version 18.8.0) [11].

## 3. Results

### 3.1. Systematic Review

Figure 1 presents a comprehensive overview of the steps undertaken in our literature search. The initial search strategy yielded 1873 potentially relevant articles from electronic databases. After removing duplicates, 884 articles remained. Following title and abstract screening, 826 studies were deemed irrelevant and excluded. The remaining 58 studies underwent full-text evaluation of their full text to determine eligibility. After full-text screening, 17 articles were eliminated for several reasons, as detailed in Figure 1. A total of 41 studies fulfilled the inclusion criteria and were subsequently incorporated into the systematic review.

#### 3.1.1. Characteristics of Included Studies

The characteristics of the selected articles are summarized in Appendix A. Forty-one studies fulfilled the eligibility criteria, with 270,067 research participants in total. Among this study population, 237,543 (87.96%) individuals were diagnosed with T2D, 15,173 (5.62%) with T1D, 140 (0.05%) with latent autoimmune diabetes in adults (LADA), and 17,211 (6.37%) with undifferentiated diabetes. The analyzed studies were sourced from multiple countries, including China [12,13,14,15,16,17,18,19,20,21,22,23,24,25,26], Germany [27], Japan [28,29], India [30,31,32], the United States of America (USA) [33,34,35,36], Ghana [37], Denmark [38], the Republic of Singapore [39], the USA/Qatar [40], Thailand [41], the Netherlands [42], the Republic of Korea [43], the United Arab Emirates [44], the Netherlands/Scotland [45], Sweden [46], Montenegro [47,48], Germany/Austria [49], and Spain [50].

The sample size varied considerably across studies, ranging from 95 to 114,231 patients. The largest sample size, consisting of data from 114,231 people with newly diagnosed T2D from the National Diabetes Register of Sweden, was incorporated into the research conducted by Lugner et al. [46]. The study conducted at the Primary Health Care Centre in Podgorica, Montenegro, had the smallest sample size, with only 95 participants [47]. The disease duration ranged from less than one year [22,27] to thirty-eight years or more [36]. In most studies, the diagnosis of diabetes was established based on the criteria of the World Health Organization (WHO) [51,52] and the American Diabetes Association (ADA) [53,54,55,56,57]. In some studies, diabetes was diagnosed using certain laboratory parameters [13,15,19,20] and assessment of specific health conditions and/or administration of hypoglycemic pharmacotherapy [26,28,30,36,37,45], as well as on the basis of the ICD codes [29,33,46,48,50]. One study relied on self-reported information to ascertain the presence of diabetes [34].

Data were obtained from multiple healthcare facilities and other sources, including university hospitals, tertiary care centers, outpatient clinics, primary care medical facility databases, national research centers, disease or clinical registries, EHR, and surveys. The study designs of the reviewed publications were as follows: twenty-two cohort studies, 14 cross-sectional studies, and five studies utilizing data from RCTs. The follow-up duration in cohort studies varied from approximately two [41,49] to fourteen years [43].

#### 3.1.2. Methods for Clustering and Dimensionality Reduction Techniques

The cluster analysis methodologies employed in the selected studies are presented in Appendix A. K-means clustering was the most commonly utilized method for clustering in the reviewed publications [12,16,17,18,19,20,21,22,23,24,25,28,29,30,31,32,34,36,38,39,40,41,42,45,46]. The next most prevalent clustering techniques were hierarchical cluster analysis [33,47,49] and two-step clustering [13,14,15,26,37,48]. Several studies have employed a combination of techniques for performing cluster analysis [43,44,50,58]. K-prototype clustering is the least commonly used cluster analysis method [35]. Five studies emphasized the use of dimensionality reduction techniques as a preparatory step before conducting cluster analysis [12,16,19,21,23]. Appendix A lists the variables used in the cluster analysis.

#### 3.1.3. Quality Assessment Results

Overall, all cohort studies achieved scores above 70%, indicating moderate-to-high quality of evidence. The cross-sectional studies were assessed as having a risk of bias ranging from low to moderate. The RCTs were rated as having a low risk of bias or as research efforts with some concerns. The findings of the quality control assessment are summarized in Appendix A.

#### 3.1.4. Prevalence of Clusters

The reviewed studies identified three [13,32,39,40], four [16,17,18,19,21,22,23,31,34,38,43,58], and five [12,14,15,27,28,29,33,37,59] distinct subtypes in their approach to generating clusters similar to those initially observed in the Scandinavian population [4].

Table 2 presents the pooled prevalence estimates for clusters derived from studies that identified five clusters [12,14,15,27,28,29,33,37,59] under the REM. The studies showed significant between-study heterogeneity (I^2^ > 90%, Ph < 0.001). The SAID cluster had the lowest pooled prevalence (8%; 95% CI: 6.0–11%), while the MOD (31%; 95% CI: 23–39%) and MARD (27%; 95% CI: 21–34%) clusters had the highest prevalence. Subgroup analysis stratified by ethnic group indicated a higher prevalence of SIDD (25%; 95% CI: 16–34% vs. 11%; 95% CI: 1.0–23%), SIRD (14%; 95% CI: 10–19% vs. 10%; 95% CI: 4.0–15%), and MARD (29%; 95% CI: 23–34% vs. 22%; 95% CI: 10–33%) among individuals of Asian descent than among individuals of other ethnic groups.

Table 3 presents both the overall and subgroup analyses under the REM, providing estimates for the pooled prevalence of the four clusters identified in studies on T2D [16,17,18,19,21,22,23,31,32,38,40,43,58]. There was strong evidence of heterogeneity among the studies (I^2^ > 88%, Ph < 0.001). In the overall analysis, SIRD had the lowest pooled prevalence (17%; 95% CI: 14–19%), whereas MARD was the most frequently observed cluster (37%; 95% CI: 34–40%). The pooled prevalence of SIDD (22%; 95% CI: 18–26% vs. 9%; 95% CI: 8.0–10%) and MOD (30%; 95% CI: 25–35% vs. 26%; 95% CI: 25–28%) was notably higher in Asian populations than in individuals from other ethnicities. In contrast, SIRD and MARD were more predominant in the non-Asian groups.

#### 3.1.5. Characteristics of Clusters

Appendix A provides a summary of the key characteristics of the clusters, which align either fully or partially with those identified by Ahlqvist et al. [4]. Numerous studies have replicated the same major diabetes subtypes, which share largely similar features and phenotypic characteristics. Individuals assigned to SAID exhibited positive GADA status, relatively early onset of the disease, low or depleted insulin secretory capacity, comparatively low BMI, and poor long-term glycemic control based on HbA1c. The key distinguishing characteristic of SIDD was GADA negativity; otherwise, its clinical profile closely resembled that of SAID. The SIRD cluster was characterized by high BMI, severe insulin resistance, and elevated β-cell function. Glycemic control was relatively well managed. Individuals in the MOD cluster were distinguished by a relatively young age, high BMI, and less pronounced insulin resistance than those in the SIRD cluster. Participants with MARD differed by having older-age-onset diabetes and mild to moderate metabolic abnormalities, with no further extreme characteristics noted.

Beyond the variations in clinical characteristics across clusters, there was a notable divergence in complications, highlighting a heterogeneous risk profile among the diabetes population. A higher prevalence and risk of diabetic retinopathy (DR) were predominantly observed in the SIDD subtype [12,13,17,28,30,31,59]. Zhang et al. reported that participants assigned to the SAID cluster had a higher incidence of both DR and diabetic peripheral neuropathy (DPN) [15]. Christensen et al. noted that the MARD cluster had a higher prevalence of diabetic eye disease [38].

The SIRD subtype is associated with a wide spectrum of clinical presentations of kidney involvement. Individuals classified into the SIRD cluster were most likely to develop diabetic kidney disease (DKD) [12,14,19]. The SIRD subtype is linked to a lower median eGFR at baseline [58] and a higher risk and prevalence of chronic kidney disease (CKD) [13,17,22,39,59], macroalbuminuria [59], microalbuminuria [28], and renal dysfunction [29,38]. Notably, Song et al. found no statistically significant difference in the risk of DKD between subgroups after adjusting for HOMA2-IR [14]. In the GoDARTS cohort, patients assigned to SIRD had nearly a two-fold higher risk of developing CKD and end-stage renal disease (ESRD), while patients with MOD faced an approximately five-fold higher risk of ESRD relative to those in the mild diabetes with high high-density lipoprotein cholesterol (MDH) subgroup [45]. Individuals in the SIDD cluster also have a higher prevalence of albuminuria [17] and risk of CKD [24]. Hwang et al. found that individuals in the MARD and SIDD clusters were more likely to develop CKD, defined as at least stage 3A, than those in the SIRD subgroup [43].

Song et al. found that individuals in the MOD were at a higher risk of developing non-alcoholic fatty liver disease (NAFLD), even after adjusting for BMI and HOMA2-IR [14]. However, Wang et al. reported that participants assigned to the severe obesity-related and insulin-resistant diabetes (SOIRD) cluster were more likely to have incident metabolic-associated fatty liver disease (MAFLD), followed by those in the MOD cluster compared to the MARD subtype [20]. In a study by Hwang et al., the SIRD cluster demonstrated the highest prevalence of MAFLD, with 97% of patients affected [43]. Li and Chen observed consistent results, with the greatest prevalence of NAFLD identified among individuals with SIRD (74.5%), followed by those in MOD (72.1%) [22].

Lu et al. reported that the risk of diabetic neuropathy was overall higher in SIDD than in other clusters [33]. Xiong et al. revealed that individuals in the SIDD cluster were more prone to diabetic foot, while patients in the SIRD faced a greater risk of DPN [13].

An unfavorable cardiovascular risk profile was evident across all clusters, with no distinct patterns (SAID [15], SIDD [33,43], SIRD [12,28,39,45,59], MOD [22], MARD [24,43], SOIRD [24], and uric acid-related diabetes (UARD) [13].

Several studies have examined the therapeutic response to antidiabetic agents, suggesting that certain clusters exhibit unique patterns of treatment efficacy. Huang et al. observed that within the MARD cluster, telomere lengths (TLs) were significantly longer in the group receiving metformin than in the metformin-free group, even after adjusting for age and sex [16]. According to Pigeyre et al., glargine decreased hyperglycemia by 19%, 13%, and 9% in the SAID, SIDD, and MOD clusters, respectively, compared to standard care, with MARD serving as the reference group [59]. In a study by Zou et al., canagliflozin demonstrated the strongest glucose-lowering effect in the MOD cluster over 52–104 weeks, outperforming both sitagliptin and glimepiride [58]. In addition, the highest proportion of participants achieving the HbA1c target (<7.0%) in the MARD were those treated with sitagliptin [58]. Abdul-Ghani et al. found that Group 1, which was comparable to the SIDD, demonstrated a more favorable reduction in HbA1c over three years when treated with combination therapy, including pioglitazone and exenatide, compared to insulin therapy or the conventional treatment regimen of metformin, followed sequentially by glipizide, and glargine insulin [40].

Appendix A presents the clusters identified using alternative clustering methodologies that do not directly correspond to those described by Ahlqvist et al. [4].

Lugner et al. identified four clusters in T2D patients [46]. Individuals with a high BMI (Cluster 1) exhibited worse glycemic control and less favorable lipid profiles. Patients diagnosed at an older age (Cluster 3) tended to have more controlled blood glucose levels. Younger individuals with a high BMI (Cluster 4) were more likely to maintain close-to-target glycemic control.

Cojic et al. identified four distinct phenotypes among individuals with T2D using a clustering approach that integrated various parameters, including the vitamin D status [47]. Notably, the cluster with the highest vitamin D levels exhibited significantly lower insulin resistance than those with vitamin D deficiency or insufficiency. Additionally, this cluster demonstrated superior glycemic control compared to the clusters with vitamin D insufficiency.

Grimsmann et al. identified eight data-driven clusters of adults with T2D and T1D [49]. Four of these clusters were characterized by early onset diabetes, with a median age at diagnosis of between 40 and 50 years. Despite the shared age range, these clusters exhibited significant heterogeneity in key clinical variables, including BMI, HbA1c levels, prevalence of diabetic ketoacidosis (DKA), and β-cell antibody positivity. Two clusters were composed of individuals diagnosed in their early 60s, who generally achieved target HbA1c levels but showed notable differences in BMI and sex distribution within the clusters. The remaining two clusters represented late-onset diabetes, diagnosed in the late 60s and beyond, but were distinguished by substantial variability in the HbA1c levels.

Wang et al. stratified participants with T2D into four distinct clusters [25]. The clusters were ranked from low to high risk based on the likelihood of developing complications. Individuals in Cluster 1, labeled as low-risk, had better glycemic control, more favorable lipid metabolism control, and moderate insulin resistance that was less pronounced than in other clusters. In contrast, those in Cluster 4, designated as high-risk, exhibited very poor glycemic control, worse lipid profiles, and the greatest risk of microangiopathy, diabetic nephropathy, and DR.

In 2024, Cojic et al. revealed four unique clusters among individuals with T2D [48]. Their study specifically examined the inflammatory markers associated with each cluster, providing further insights into the role of inflammation in the pathophysiology of disease. Notably, Cluster 3, which comprised older patients with longer disease duration, suboptimal glycemic control, and the greatest prevalence of hypertension and nephropathy, exhibited the highest levels of inflammation-related markers.

Somolinos-Simon et al. identified five distinct subgroups within the T1D study population and analyzed their key differences, as well as the associated risks for diabetes-related complications over a five-year follow-up period [36]. The complications evaluated included DR, coronary artery disease (CAD), autonomic neuropathy, albuminuria, hypertension, severe hypoglycemia, DKA, and depression.

The potential for individuals with diabetes to shift between different diabetes clusters over time remains an intriguing research question. Li et al. revealed that over 28% of individuals shifted to different subgroups after just two years [45]. In the GoDARTS cohort, the Risk Assessment and Progression of Diabetes project (RHAPSODY) SIRD (RHAP-SIRD) subgroup exhibited the highest stability, with 77% of individuals remaining in the same cluster over an eight-year study period. In contrast, the RHAPSODY SIDD (RHAP-SIDD) subgroup showed the least stability, with only 8% of individuals remaining in the same cluster. There was no significant difference in the risk of macrovascular disease between individuals who transitioned from the mild subgroups (RHAPSODY mild diabetes (RHAP-MD), RHAP-MOD, and RHAP-MDH) to the severe subgroups (RHAP-SIDD and RHAP-SIRD) and those who remained in the severe subgroups over a two-year period. However, among individuals initially classified in the severe subgroup, those who remained in this category over the two years demonstrated a significantly higher risk of developing CKD than individuals who transitioned to the mild subgroups. In contrast, individuals initially classified in the mild subgroup who progressed to the severe subgroup during the two-year follow-up demonstrated an increased risk of acute myocardial infarction (AMI) and congestive heart failure (CHF) relative to those who remained in the mild subgroup.

### 3.2. Cross-Sectional Study

The research participants had a median age at diagnosis of 57.9 years (interquartile range (IQR), 48.8–64.3) and a median BMI of 30.5 kg/m^2^ (IQR, 27.5–34.0). Blood pressure measurements revealed a median SBP of 120 mm Hg (IQR, 120–130) and a median DBP of 80 mm Hg (IQR, 80–80). The median HbA1c level, an indicator of long-term glycemic control, was 7.7% (IQR, 6.8–9.5). The median TC and LDL-C, lipid metabolism markers, were 5.3 mmol/L (IQR, 4.5–6.2) and 3.4 mmol/L (IQR, 2.7–4.0), respectively. The median eGFR was 96.7 mL/min/1.73 m^2^ (IQR, 79.3–106.9). Women comprised 56.5% of the participants.

In our study based on Kazakhstani data, the participants were categorized into five distinct clusters. The characteristics of each cluster are summarized in Table 4 and Figure 2. Cluster 1 comprised 176 (31.5%) older individuals who exhibited suboptimal glycemic control and a relatively favorable lipid profile with borderline optimal TC and LDL-C levels. Cluster 2 included 83 (14.9%) individuals characterized by late-onset diabetes, higher BMI corresponding to Obesity Class I, elevated SBP and DBP, suboptimal glycemic control, mild lipid abnormalities with TC and LDL-C in the borderline-high range, and lower eGFR. Cluster 3 consisted of 98 (17.6%) individuals who were diagnosed at a younger age and demonstrated a lower BMI within the overweight range, suboptimal glycemic control, and better lipid metabolism control, with lower levels of TC and LDL-C. In Cluster 4, 110 (19.7%) individuals presented with high BMI levels consistent with Obesity Class I and suboptimal glycemic control, along with a more unfavorable lipid profile, marked by higher TC and LDL-C levels. Finally, Cluster 5, comprising 91 (16.3%) individuals, featured overweight status, very poor glycemic control, and mild dyslipidemia with borderline-high levels of TC and LDL-C.

## 4. Discussion

Our systematic review demonstrates that unsupervised cluster analysis holds significant potential for effectively capturing the complex heterogeneity within the diabetes population and identifying distinct and homogeneous clusters characterized by pathophysiologically feasible factors and unique clinical profiles. Another key observation was the remarkable reproducibility of the five clusters across a diverse range of participants, encompassing various patient origins and ethnic backgrounds. Although the clusters shared similar phenotypic characteristics and a common risk profile for complications, there was substantial variability in the distribution of these clusters and the central tendencies of the key variables. Studies utilizing alternative clustering techniques and algorithmic model variables have revealed additional clinically relevant clusters. In general, cluster analysis can be employed to empirically explore phenotypic traits and diagnostic differences across homogenous and clinically meaningful diabetes subtypes, although it is subject to certain limitations.

A distinct issue to consider is the potential influence of participants’ ethnic backgrounds on the distribution and composition of the clusters. Conflicting evidence exists regarding the differences in the prevalence of SIDD between individuals of European and Asian origin [60]. Our systematic review highlights substantial variability in the prevalence of SIDD across different populations. Notably, Asian groups, particularly East Asians, who were represented in 16 of the twenty-two studies, exhibited higher SIDD prevalence rates compared to their non-Asian counterparts. The present findings comply with the conclusions drawn in the previously published research conducted by Varghese and Narayan [61]. East Asians generally demonstrate a lower endogenous capacity for insulin secretion [62,63,64,65], and T2D tends to develop at a lower average BMI than that of individuals of European ancestry [66]. Overall, in lean individuals with T2D who do not have severely impaired insulin sensitivity, β-cell exhaustion is thought to progress with age and prolonged exposure to glycemic stress, often exacerbated by poor dietary habits [60]. Over several decades, this ongoing metabolic strain gradually reduces the pancreas’s intrinsic ability to secrete insulin, eventually leading to the onset of T2D [60,67].

Based on our findings, the prevalence of SIRD across Asian and other demographic groups appears inconsistent, leading to uncertainty regarding its ethnic-specific prevalence. Studies have consistently shown that individuals of South Asian heritage exhibit a greater predisposition to insulin resistance than their White counterparts [68,69,70,71,72]. A recent study further revealed that non-Hispanic Whites and African Americans experience lower insulin resistance levels than South and East Asian individuals [73]. One possible explanation for this disparity is that individuals of South Asian descent may have a greater biological propensity to accumulate fat, predominantly in the abdominal region, with a lower overall BMI [2,62]. Generally, individuals with ectopic fat accumulation in the liver and skeletal muscles, resulting from a diminished lipogenic capacity of subcutaneous fat, are more prone to developing insulin resistance, which is a key factor underlying the onset of T2D [60,74]. Indeed, recent research has demonstrated that the South Asian-American population has lower β-cell function, increased insulin resistance, and less favorable body composition, characterized by higher levels of liver and intermuscular fat and lower muscle mass, compared to four other racial and ethnic groups in the United States [75]. However, contrary to the prevailing belief, Narayan et al. argued that South Asians exhibit lower levels of insulin resistance and greater insulin deficiency relative to both U.S. Black and White populations [76]. Further research is crucial to elucidate the predominant pathophysiological mechanisms driving the high prevalence of T2D in different ethnic groups.

In addition to the observed differences in clinical characteristics among the identified clusters, cluster analysis also revealed distinct cluster-specific risk profiles for complications. The higher prevalence and increased risk of DKD are predominantly associated with the SIRD subtype. Insulin resistance is a key pathogenic driver of various metabolic diseases, including T2D [77,78]. A potential pathogenetic pathway linking insulin resistance to renal dysfunction involves a multifactorial interplay between metabolic syndrome, adipocytokine dysregulation, hyperinsulinemia, and chronic low-grade inflammation [79]. These interconnected factors collectively promote renal injury by disrupting various biochemical and molecular pathways, particularly the insulin-signaling pathway in the kidneys [79], which plays a pivotal role in the pathophysiology of renal dysfunction [80,81]. Insulin signaling is crucial for several key renal processes, including the maintenance of glomerular function [80,82], renal gluconeogenesis [83], tubular transport [84,85], and vascular regulation [80]. Disruption of insulin action may impair renal hemodynamics [86,87], compromise podocyte viability [88,89,90] and negatively affect tubular function [80,91,92].

SIDD is a high-risk group for the development of DR, potentially resulting from sustained and inadequate glycemic control, which accelerates microvascular damage in the retinal tissue. Multiple lines of evidence have consistently demonstrated that the development and progression of DR are strongly correlated with prolonged hyperglycemia [93,94,95,96]. Moreover, a growing body of research has established a significant relationship between long-term variability in HbA1c levels and both the onset [97] and progression of DR [98,99], underscoring the importance of glycemic stability in mitigating the risk of retinal damage.

Cluster analysis indicated that individuals within the SIRD cluster are at a significantly higher risk of developing NAFLD, now termed MAFLD, with a similar trend observed in the SOIRD and MOD clusters. T2D and NAFLD share a complex bi-directional relationship, with each driving the onset and progression of the other [100,101]. Beyond the well-established relationship between insulin resistance and an increased incidence of CVD [102,103], NAFLD has been independently associated with a heightened long-term risk of both fatal and non-fatal CVD events [104,105] and an increased risk of CKD [105,106].

The SAID, SIDD, and SIRD clusters exhibited a notably high risk of diabetic neuropathy, driven by overlapping pathophysiological mechanisms. Chronic hyperglycemia, insulin resistance, and dyslipidemia activate several key metabolic pathways, including the polyol, hexosamine, and protein kinase C (PKC) pathways [107,108]. Together, these processes ultimately foster mitochondrial dysfunction, oxidative stress, inflammation, and alterations in gene expression, leading to intracellular damage, nerve dysfunction, and neuronal cell death [107,108].

It remains uncertain whether clustering can detect subgroups of individuals who would experience greater therapeutic benefits from specific classes of medications used to manage hyperglycemia. Pigeyre et al. demonstrated that insulin glargine exhibited superior efficacy in reducing elevated blood glucose levels in the SIDD cluster when compared to standard care [59]. This finding underscores the benefit of initiating insulin therapy early in this specific patient cohort, suggesting that early insulin intervention may provide enhanced control of hyperglycemia in individuals with SIDD. Canagliflozin, a sodium-glucose co-transporter 2 (SGLT2) inhibitor, demonstrated superior glucose-lowering efficacy compared with sitagliptin, a dipeptidyl peptidase-4 (DPP-4) inhibitor, and glimepiride, a sulfonylurea, in the MOD cluster [58]. The enhanced efficacy of canagliflozin is likely attributable to its distinct mechanism of action, which involves the inhibition of renal glucose reabsorption, thereby enhancing urinary glucose excretion [109]. Additionally, canagliflozin promotes weight loss and exerts beneficial effects on both cardiovascular and renal outcomes [110]. Zou et al. further demonstrated that participants in the MARD cluster treated with sitagliptin had the highest proportion of individuals achieving the target HbA1c level [58]. This may be attributed to the drug’s mechanism of action, which enhances insulin secretion and thereby improves glucose homeostasis [111]. Moreover, its favorable safety profile, characterized by a neutral effect on body weight and a lack of increased risk of hypoglycemia, distinguishes sitagliptin from other antidiabetic therapies [112]. Abdul-Ghani et al. demonstrated that patients in a subgroup comparable to the SIDD cluster experienced a more favorable reduction in HbA1c when treated with combination therapy, including pioglitazone and exenatide, compared to those receiving insulin therapy or the conventional regimen of metformin, glipizide, and glargine insulin [40]. In addition to augmenting insulin sensitivity through the activation of peroxisome proliferator-activated receptor gamma (PPAR-γ) [113], pioglitazone also contributes to the improvement of β-cell function [114,115,116,117]. Exenatide, a glucagon-like peptide-1 receptor agonist (GLP-1 RA), exerts its effects through the direct stimulation of insulin secretion, suppression of glucagon release [118,119], and promotion of β-cell proliferation [120]. The results of this study indicated that, in specific patient subsets, agents that preserve β-cell function may offer superior efficacy in enhancing insulin sensitivity compared to insulin replacement therapy. These findings partially contradict those reported by Pigeyre et al. [59] and underscore the necessity for further research to evaluate the therapeutic response of diabetes clusters to glucose-lowering treatment.

Other studies employing alternative methodologies have identified distinct diabetes clusters that only partially align with those proposed by Ahlqvist et al. [4]. These variations indicate the complexity of diabetes as a heterogeneous disease and underscore the need for further refinement in subgroup identification. Future research is required to explore whether these alternative clusters represent truly distinct pathophysiological mechanisms or reflect methodological differences in clustering approaches.

Diabetes encompasses a broad spectrum of heterogeneity, with individuals falling into distinct subgroups that may be driven by underlying genetic factors and molecular signatures. However, pharmacological treatments, lifestyle interventions, and personalized care plans may shift individuals from one subgroup to another. This dynamic movement may be further influenced by the patient’s response to treatment and disease progression, highlighting the need for continuous monitoring and reassessment of diabetes subtypes to optimize management strategies.

The findings of our cross-sectional study based on Kazakhstani data highlighted distinct subgroups of individuals with diabetes, each with unique characteristics in terms of glycemic control, lipid metabolism, blood pressure regulation, renal function, and weight status. Notably, certain subgroups demonstrated moderate-to-severe metabolic disturbances, higher blood pressure, and pronounced renal impairment. In contrast, the other subgroups exhibited a relatively favorable clinical profile. Clusters characterized by more adverse indicators likely represent high-risk populations. The clustering approach used in this study has several critical clinical and research implications. The variation in HbA1c levels suggests that different clusters may benefit from personalized treatment strategies rather than standardized approaches. Differences in eGFR across clusters reinforce the importance of routine kidney function monitoring to detect renal impairment and initiate nephroprotective strategies. Cardiovascular risk assessment should be prioritized in clusters with dyslipidemia, hypertension, and kidney dysfunction. Clusters with higher BMI require targeted weight-loss strategies, while those with lower BMI may require alternative metabolic interventions to optimize insulin sensitivity. In essence, the identified variations in health markers point to key areas that require targeted interventions to improve patient outcomes.

Speculatively, among the five distinct clusters identified in our study, several showed partial similarities to those originally described by Ahlqvist et al. [4]. Cluster 1, comprising older individuals with suboptimal glycemic control and a relatively favorable lipid profile, shared features with the MARD cluster. Cluster 2 included individuals characterized by late-onset diabetes, higher BMI, elevated blood pressure, suboptimal glycemic control, and lower eGFR, partially aligning with the SIRD and, to a lesser extent, with the MOD cluster. Cluster 5, composed of individuals with relatively lower BMI and very poor glycemic control, could correspond to the SIDD cluster. However, these similarities should be interpreted with caution, as our cluster analysis did not include variables reflecting insulin secretion capacity or insulin resistance.

Since fasting insulin, C-peptide, and GADA levels are not routinely measured in our clinical settings, our clustering approach may be more applicable to real-world healthcare environments. However, given the cross-sectional design of our study, the identified clusters reflect phenotypic patterns observed at a single point in time, limiting our ability to assess longitudinal changes, disease progression, or the causality between clinical features and outcomes. Despite this limitation, the use of real-world clinical data provides valuable insights into the heterogeneity of diabetes within the studied population. To build on these findings, longitudinal follow-up studies are essential to identify the development of diabetes-related complications specific to each cluster and evaluate the impact of diabetes progression on patient clustering. Future research may also offer valuable insights into how different treatment strategies affect distinct patient subgroups.

This systematic review has several limitations. First, there was diversity in the methodologies used across the included studies. The inclusion of studies employing various diabetes clustering techniques introduces significant methodological heterogeneity, limiting the ability to draw consistent conclusions and affecting the generalizability of the results. Second, the inclusion of studies using different datasets with varying sample sizes, demographics, and clinical settings may introduce bias, as the identified clusters may be context-dependent. Studies also vary in terms of study design, follow-up duration, population characteristics, and diagnostic criteria, which further complicates the synthesis. Finally, the observed high between-study heterogeneity could potentially obscure the true prevalence of the clusters.

## 5. Conclusions

In conclusion, unsupervised cluster analysis offers substantial promise for identifying distinct subgroups of individuals with diabetes, providing valuable insights into disease progression and varying risks of diabetes-related complications. This technique has the potential to enhance personalized treatment strategies by distinguishing differential treatment responses, ultimately paving the way for more targeted and proactive interventions. Nevertheless, further research is needed to determine the most effective clustering algorithms and variable selection criteria, which remain key challenges in this field. Integrating these approaches with clinical data is essential to enhance their applicability in real-world healthcare settings and ultimately improve patient outcomes.

## Figures and Tables

**Figure 1 jcm-14-03588-f001:**
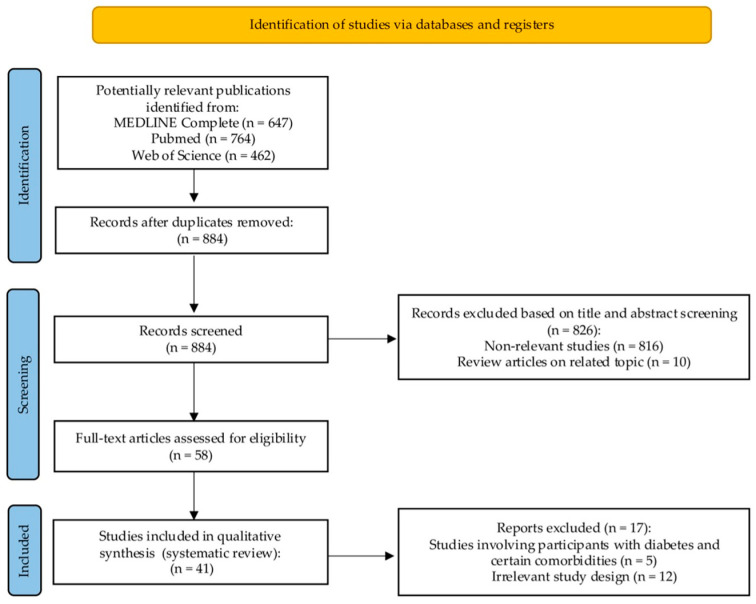
PRISMA flow diagram presenting the results of the literature search and study selection process.

**Figure 2 jcm-14-03588-f002:**
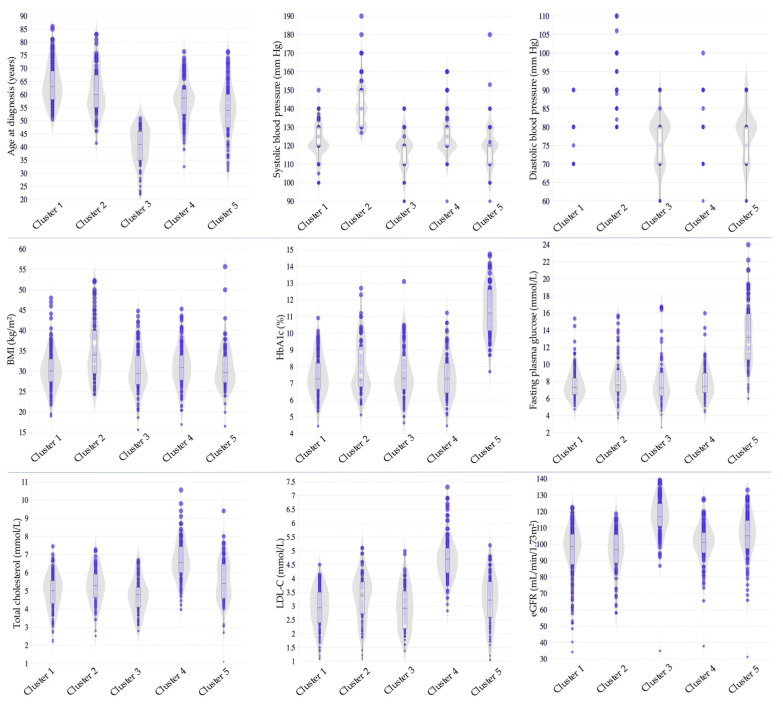
Cluster profiling based on the Kazakhstani data. BMI, body mass index; eGFR, estimated glomerular filtration rate; HbA1c, hemoglobin A1c; LDL-C, low-density lipoprotein cholesterol.

**Table 1 jcm-14-03588-t001:** Search strategies for electronic bibliographic databases.

MEDLINE Complete	Pubmed	Web of Science
(MH “Diabetes Mellitus+” OR “Diabetes Mellitus” OR “Diabetes Mellitus Type 2” OR “Diabetes Mellitus Type 1” OR Diabetes) AND (MH “Cluster Analysis+” OR “Cluster analysis” OR Cluster)	(“Diabetes Mellitus” [Mesh] OR “Diabetes Mellitus” OR “Diabetes Mellitus Type 2” OR “Diabetes Mellitus Type 1” OR Diabetes) AND (“Cluster Analysis” [Mesh] OR “Cluster analysis”)	(ALL = (diabetes)) AND ALL = (“cluster analysis”)

**Table 2 jcm-14-03588-t002:** The pooled prevalence of clusters in studies identified five clusters.

Cluster	No. of Studies *	Sample Size	Prevalence (%)	95% CI (%)
Overall				
SAID	9	775	8	6–11
SIDD	9	3190	20	13–27
SIRD	9	1849	13	1–15
MOD	9	3659	31	23–39
MARD	9	4431	27	21–34
Asian population				
SAID	5	308	7	4–10
SIDD	5	1264	25	16–34
SIRD	5	699	14	10–19
MOD	5	1157	24	18–30
MARD	5	1310	29	23–34
Other populations				
SAID	3	226	12	5–18
SIDD	3	332	11	1–23
SIRD	3	236	10	4–15
MOD	3	907	46	19–72
MARD	3	448	22	10–33

MARD, mild age-related diabetes; MOD, mild obesity-related diabetes; SAID, severe autoimmune diabetes; SIDD, severe insulin-deficient diabetes; SIRD, severe insulin-resistant diabetes. * The study by Pigeyre et al. [59] was excluded from the subgroup analysis due to its multiethnic composition.

**Table 3 jcm-14-03588-t003:** The pooled prevalence of clusters in studies on type 2 diabetes (T2D) identified up to four clusters.

Cluster	No. of Studies *	Sample Size	Prevalence (%)	95% CI (%)
Overall				
SIDD	13	2748	22	18–26
SIRD	13	2817	17	14–19
MOD	12	3995	29	26–33
MARD	12	5915	37	34–40
Asian population				
SIDD	11	1102	22	18–26
SIRD	11	918	16	13–19
MOD	10	1623	30	25–35
MARD	10	1838	37	33–40
Other populations				
SIDD	2	344	9	8–10
SIRD	2	853	23	22–25
MOD	1	923	26	25–28
MARD	2	1509	42	40–43

MARD, mild age-related diabetes; MOD, mild obesity-related diabetes; SIDD, severe insulin-deficient diabetes; SIRD, severe insulin-resistant diabetes. * The study by Zou et al. [58] was excluded from the subgroup analysis due to its multiethnic composition. The study by Abdul-Ghani et al. [40] comprised Asian and other populations, each of which was assigned to the appropriate subgroup analysis. Some studies did not identify MARD [32] or MOD [40].

**Table 4 jcm-14-03588-t004:** Cluster profiling based on Kazakhstani data.

	Cluster 1	Cluster 2	Cluster 3	Cluster 4	Cluster 5
n (%)	176 (31.5)	83 (14.9)	98 (17.6)	110 (19.7)	91 (16.3)
Age at diagnosis (years)	62.9 (58.0–69.0)	60.0 (55.1–67.5)	41.0 (35.0–46.0)	58.8 (52.6–62.0)	54.0 (47.5–60.2)
Systolic blood pressure (mm Hg)	120 (120–130)	140 (130–150)	120 (110–120)	120 (120–130)	120 (110–121)
Diastolic blood pressure (mm Hg)	80 (80–80)	90 (90–90)	80 (70–80)	80 (80–80)	80 (70–80)
BMI (kg/m^2^)	30.1 (27.5–33.1)	32.7 (29.5–35.7)	29.4 (26.8–33.8)	30.9 (27.8–33.9)	29.7 (27.2–33.7)
HbA1c (%)	7.3 (6.7–8.2)	7.2 (6.8–9.2)	7.3 (6.7–8.6)	7.3 (6.4–8.2)	11.2 (10.1–12.6)
Fasting plasma glucose (mmol/L)	7.3 (6.5–8.4)	7.6 (6.8–9.4)	7.2 (6.3–9.0)	7.4 (6.7–8.9)	13.6 (10.5–15.9)
Total cholesterol (mmol/L)	5.0 (4.3–5.6)	5.3 (4.6–5.9)	4.8 (4.1–5.2)	6.6 (6.0–7.4)	5.4 (4.6–6.5)
LDL-C (mmol/L)	2.9 (2.4–3.5)	3.4 (2.7–3.9)	2.9 (2.2–3.5)	4.7 (4.2–5.1)	3.2 (2.6–3.9)
eGFR (mL/min/1.73 m^2^)	89.8 (74.7–102.4)	84.7 (71.3–100)	110.5 (93.9 –120.2)	97 (81.6–102.9)	102.4 (84.2–109.4)

Data are presented as median and interquartile range (IQR). BMI, body mass index; eGFR, estimated glomerular filtration rate; HbA1c, hemoglobin A1c; LDL-C, low-density lipoprotein cholesterol.

## Data Availability

The original contributions presented in this study are included in the article and the Appendix A. Further inquiries should be directed to the corresponding author.

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
