# Peer review of "Cluster Analysis in Diabetes Research: A Systematic Review Enhanced by a Cross-Sectional Study"

_jcm, 2025, doi:10.3390/jcm14103588_

Round 1
Reviewer 1 Report
Comments and Suggestions for Authors
In this research, the authors reported systemic review of studies for clustering of diabetes based on clinical information. They presented results of meta-analysis of prevalence for each clusters defined by the clustering, and analyzed their own data with the clustering and identified 5 clusters having different clincial characteristics.
The results were interesting, but I'd like to address two issues.
First, in the Table 4 and 5, they summarized the characteristics of clusters with the descriptions. I believe that the current style would be difficult to recognize the characteristics. Instead, I'd like to suggest authors to move the current Tables to the Supplementary Materials, and the Tables should be re-orgarnized into a more concise form. For example, all the characteristics are assigned to columns, and positive or negative markings for the variables can be made in each cluster.
Second, although the authors summarized characteristics of the clusters defined by their own data, I think it would be better to compare the results with the previoous clusters. Especially, it seems to be important to link the clusters of this research to the previous clusters, which can support the validity of the 5 clusters (SAID, SIDD, SIRD, MOD, MARD).
Author Response
Manuscript ID: jcm-3592022
Binura Taurbekova
MD, Nephrologist, M.S. in Molecular Medicine, Ph.D. in Global Health
Nazarbayev University School of Medicine
5/1 Kerey and Zhanibek Khandar Str.,
Astana, Republic of Kazakhstan
E-mail: binura.taurbekova@nu.edu.kz
Date: May 3, 2025
On behalf of my colleagues and co-authors, I would like to express our sincere gratitude to the reviewers for their thorough evaluation and constructive feedback. We have carefully considered all reviewers’ suggestions and incorporated the necessary revisions into the manuscript. All changes have been highlighted in yellow for clarity. Below, we provide a point-by-point response to each comment.
Reviewer 1:
Comment 1: First, in the Table 4 and 5, they summarized the characteristics of clusters with the descriptions. I believe that the current style would be difficult to recognize the characteristics. Instead, I'd like to suggest authors to move the current Tables to the Supplementary Materials, and the Tables should be re-orgarnized into a more concise form. For example, all the characteristics are assigned to columns, and positive or negative markings for the variables can be made in each cluster.
Response 1: Thank you for your valuable suggestion. In response, we have moved the original Tables 4 and 5 to the Supplementary Materials and replaced them with revised versions (Tables S7 and S8), presented in a more concise and structured format as recommended. The main findings derived from these tables are reflected in the main text of the manuscript (Lines 331-343, 397-436).
Comment 2: Second, although the authors summarized characteristics of the clusters defined by their own data, I think it would be better to compare the results with the previous clusters. Especially, it seems to be important to link the clusters of this research to the previous clusters, which can support the validity of the 5 clusters (SAID, SIDD, SIRD, MOD, MARD).
Response 2: We appreciate the reviewer’s thoughtful suggestion. However, fasting insulin, C-peptide, and GADA are not routinely measured in our clinical settings. This limits the ability to calculate HOMA2-IR/HOMA-IR, HOMA2-B/HOMA-B, and to determine antibody status. As a result, defining clusters such as SIDD, SIRD, and SAID in our dataset is challenging. To address this, we selected clinically accessible variables that reflect real-world data availability. This enhances the applicability of our clustering approach to everyday healthcare practice. We noted this in the Discussion section, where we highlighted the relevance of our variable selection to routine clinical practice (Lines 649-656). Nevertheless, we addressed the reviewer’s concern by speculatively comparing our identified clusters with the 5 clusters in the Discussion section (Lines 639-648).
Reviewer 2 Report
Comments and Suggestions for Authors
This systematic review focuses on efficacy of unsupervised cluster analysis in identifying diabetes phenotypes along with diabetes -related complication assessment capabilities and treatment approaches. The authors have conducted the review in depth and tried to provide evidence on this area. Please consider the following minor suggestions to improve the impact of the article.
- Abstract: Line 38 – Results representation is vague. Please incorporate key findings here that you discussed in the paper – for example pooled prevalence of cluster compared between populations and brief results of the cross-sectional studies. Statements as is very generic - for example line 42-43 – no specific traits or cluster characteristics or clinical variables are listed. Results need to be a bit more specific.
- Line 80-81 “Overall, five…. different ethnic group”. Would recommend describing the characteristics of these clusters. Line 89 says your review is extending previous research efforts. So, it is important in line 80-81, that you give readers a high-level understanding of what previous cluster traits were.
- Line 167 (2.2.1 study population and design) – this does not highlight any subject characteristics, inclusion exclusion criteria. In line 94 it is mentioned that “…with a unique genetic profile…” what profile is this? Please clarify in the methods section.
- Line 274 to 279 – Briefly identify the characteristics of these clusters either in methods or here. What defines MARD, SAID SIDD etc.
- Line 282 – Table 1 number of studies does not tally unless there is some overlap in the studies with same study falling user 2 or 3 clusters. As is total studies show 45 whereas only 41 studies were included in the analysis. Same goes with split in Asian Vs other – 5 + 3 = 8, but your overall study shows 9. If there is overlap, then all clusters show cross talk or overlap? Is there scope for a 6th cluster? Please clarify.
- Section 3.2: Again, cluster characteristics is vague. What were the key variables and variable values that defines cluster 1-5. Please clarify here in results or in methods section. The section as is showing random cluster formation.
Author Response
Manuscript ID: jcm-3592022
Binura Taurbekova
MD, Nephrologist, M.S. in Molecular Medicine, Ph.D. in Global Health
Nazarbayev University School of Medicine
5/1 Kerey and Zhanibek Khandar Str.,
Astana, Republic of Kazakhstan
E-mail: binura.taurbekova@nu.edu.kz
Date: May 3, 2025
On behalf of my colleagues and co-authors, I would like to express our sincere gratitude to the reviewers for their thorough evaluation and constructive feedback. We have carefully considered all reviewers’ suggestions and incorporated the necessary revisions into the manuscript. All changes have been highlighted in yellow for clarity. Below, we provide a point-by-point response to each comment.
Reviewer 2:
Comment 1: Abstract: Line 38 – Results representation is vague. Please incorporate key findings here that you discussed in the paper – for example pooled prevalence of cluster compared between populations and brief results of the cross-sectional studies. Statements as is very generic - for example line 42-43 – no specific traits or cluster characteristics or clinical variables are listed. Results need to be a bit more specific.
Response 1: We thank the reviewer for this valuable comment. In response, we have revised the Results section of the Abstract to provide more specific information. We have reported the prevalence of clusters across populations, highlighted key findings regarding the complication risk profile, and briefly summarized the main results of the cross-sectional study.
Comment 2: Line 80-81 “Overall, five…. different ethnic group”. Would recommend describing the characteristics of these clusters. Line 89 says your review is extending previous research efforts. So, it is important in line 80-81, that you give readers a high-level understanding of what previous cluster traits were.
Response 2: We thank the reviewer for this helpful suggestion. In response, we have added a description of the characteristics of the five clusters (Lines 80–86), providing readers with a clearer understanding of the traits typically associated with these clusters.
Comment 3: Line 167 (2.2.1 study population and design) – this does not highlight any subject characteristics, inclusion exclusion criteria. In line 94 it is mentioned that “…with a unique genetic profile…” what profile is this? Please clarify in the methods section.
Response 3: We thank the reviewer for this important comment. In response, we have revised the Methods section to include additional details on the study population, including subject characteristics, and inclusion and exclusion criteria (Lines 188-191, 196-199). Furthermore, we clarified the reference to the “unique genetic profile” mentioned in Line 107 by citing findings from a recent whole-genome sequencing study of ethnic Kazakhs and briefly summarizing its relevance in the context of our study (Lines 94-101).
Comment 4: Line 274 to 279 – Briefly identify the characteristics of these clusters either in methods or here. What defines MARD, SAID SIDD etc.
Response 4: We thank the reviewer for the suggestion. The characteristics defining the clusters, including MARD, SAID, SIDD, MOD, and SIRD, are described in the relevant paragraph of the Results section (3.1.4. Characteristics of Clusters; Lines 331–343). Additionally, detailed characteristics of each cluster are presented in Supplementary Table S7.
Comment 5: Line 282 – Table 1 number of studies does not tally unless there is some overlap in the studies with same study falling user 2 or 3 clusters. As is total studies show 45 whereas only 41 studies were included in the analysis. Same goes with split in Asian Vs other – 5 + 3 = 8, but your overall study shows 9. If there is overlap, then all clusters show cross talk or overlap? Is there scope for a 6th cluster? Please clarify.
Response 5: We thank the reviewer for this detailed observation. To clarify:
- The total number of studies included in our systematic review is 41.
- Table 2 (Line 303) presents the prevalence of five clusters (SAID, SIDD, SIRD, MARD, and MOD). Data from 9 studies [12, 14, 15, 27-29, 33, 37, 59] that identified all five clusters were used to calculate the overall prevalence of the five clusters. For the subgroup analysis based on ethnicity, we included 5 studies involving Asian populations and 3 studies involving other populations. One study, Pigeyre et al. [59] was excluded from the subgroup analysis due to its multiethnic composition (578 clinical sites across 40 countries) and is marked with an asterisk (*).
- Table 3 (Line 319) presents the prevalence of up to four clusters (SIDD, SIRD, MOD, MARD) in other studies focused on type 2 diabetes [16–19, 21–23, 31, 32, 38, 40, 43, 58]. The asterisk (*) clarifies the nuances of the subgroup analysis calculation based on the following: (1) Zou et al. [58] was excluded from the subgroup analysis due to its multiethnic composition; (2) The study by Abdul-Ghani et al. [40] comprised both Asian and other populations; each was assigned to the appropriate subgroup analysis; (3) Some studies did not identify MARD [32] and MOD [40] and were therefore not included in the calculation of MARD and MOD prevalence.
Comment 6: Section 3.2: Again, cluster characteristics is vague. What were the key variables and variable values that defines cluster 1-5. Please clarify here in results or in methods section. The section as is showing random cluster formation.
Response 6: We thank the reviewer for this valuable comment. In response, we have clarified the key variables and corresponding values that define each of the five clusters in the Results section (Lines 466–480), based on detailed values presented in Table 4 and Figure 2. Additionally, we have linked our clusters to previously reported subtypes to support the validity of our clustering approach and to demonstrate that the clusters are clinically meaningful rather than random formations (Lines 622-648).